# Modeling Activity-Time to Build Realistic Plannings in Population Synthesis in a Suburban Area

Younes Delhoum [ID], Rachid Belaroussi *[ID], Francis Dupin [ID] and Mahdi Zargayouna [ID]

COSYS-GRETTIA, Université Gustave Eiffel, F-77447 Marne-la-Vallée, France;
younes.delhoum@univ-eiffel.fr (Y.D.); francis.dupin@univ-eiffel.fr (F.D.); mahdi.zargayouna@univ-eiffel.fr (M.Z.)
* Correspondence: rachid.belaroussi@univ-eiffel.fr

**Abstract:** In their daily activity planning, travelers always considers time and space constraints such as working or education hours and distances to facilities that can restrict the location and time-of-day choices of other activities. In the field of population synthesis, current demand models lack dynamic consistency and often fail to capture the angle of activity choices at different times of the day. This article presents a method for synthetic population generation with a focus on activity-time choice. Activity-time choice consists mainly in the activity's starting time and its duration, and we consider daily planning with some mandatory home-based activity: the chain of other subsequent activities a traveler can participate in depends on their possible end-time and duration as well as the travel distance from one another and opening hours of commodities. We are interested in a suburban area with sparse data available on population, where a discrete choice model based on utilities cannot be implemented due to the lack of microeconomic data. Our method applies activity-hours distributions extracted from the public census, with a limited corpus, to draw the time of a potential next activity based on the end-time of the previous one, predicted travel times, and the successor activities the agent wants to participate in during the day. We show that our method is able to construct plannings for 126k agents over five municipalities, with chains of activity made of work, education, shopping, leisure, restaurant and kindergarten, which fit adequately real-world time distributions.

**Keywords:** activity planning; time-choice; MATSim (Multi-Agent Transport Simulation); population synthesis

## 1. Introduction

Large-scale and complex mobility systems can be represented by simulating the behaviors and interactions of self-interested "agents". The simulation of mobility is particularly suitable for an agent-based design. Indeed, the objective in these simulations is to take into account human behaviors interacting in an open, dynamic, and complex environment [1]. This multi-agent paradigm provides a high level of details and allows representing non-linear phenomena and patterns that would be difficult to tackle with analytical approaches [2]. Among multi-agent models, there is a class called activity-based models, which specifically address the need for realistic representation of travel demand and the human behavior in a mobility context. The activity-based modeling (ABM) framework is capable of evaluating travel demand and transportation supply management strategies, such as road pricing and behavior modification programs (flexible scheduling, ride-sharing), in a more efficient way than the previous generation of aggregate flow models, which generally focus on evaluating network capacity improvement.

Activity-based models describe in greater detail than traditional models the travel patterns related to activities of households throughout a period of time and for a specific studied region. In these models, detailed descriptions of individuals and the households in terms of socio-economic attributes is key to guaranteeing the behavioral aspects implied by the models [3]. In this context, population synthesis is the sub-model that ensures the

specification of an estimated population, whose complex configuration needs to be properly captured. A synthetic population is a representation of the actual population in a certain region and focuses only on a set of attributes of the actual population. This population exists only artificially (statistically) and is an approximation of the real population of the studied area [4].

The objective of an activity-based model is to predict, for each individual of the synthetic population, which activity is generated (activity-type), when it is executed (start-time), for how long (duration), where the activity takes part (location-choice), and the transport mode used (mode-choice). Most models struggle with the dynamic consistency due to the difficulty to capture the trade-off between activities scheduled throughout a traveller's daily path [5]. The choices depend on each other: the mode used depends on the activity location, the time-choice depends on the activity-type (typical duration, time-windows), and in the case of type of activity pattern (for a day, a week, etc.), the activity order in the activity plan influences the choice of activity time and its duration. Some other dependencies (constraints) can be included, such as the travel time, which depend mainly on the mode-choice and location-choice.

In this paper, we are interested in the activity time-choice. Given an activity plan, several approaches can be used to set the time-choice (mainly as part of the scheduling process). The objective is to find a realistic pattern for these activities by considering the activity-choice, location-choice, and mode-choice. It involves the introduction of several temporal-constraints related to the activity-duration and the activity-time distribution. The output of the proposed approach is an activity-plan with realistic times and correlated to the pre-defined time-distributions of the activities.

Following their decision-making process (behavior-choice), activity-based models can be divided into two classes: econometric activity-based models and rule-based models. The econometric models use mainly utility maximization-based equations, with the aim to predict the probability of decision outcomes (based on the relationship between the travel and activity attributes). Several econometric models were developed, such as CEMDAP [6] and MORPC [7]. A rule-based model (also called computational process model), is a computer program built on a set of rules (a production system model). Each rule has a format of (if "condition" then "action"), it is applied when its condition is met. This approach can capture the schedules constraints directly. Several rule-based models were developed, such as SCHEDULER [8], ALBATROSS [9], TASHA [10].

The remainder of this paper is structured as follows. In Section 2, we describe previous works. In Section 3, we precisely define the problem that we are addressing. In Section 4, we introduce the proposed methodology to set the time-choice. In Section 5, we describe the travel demand. In Section 6, we describe the implementation of the time-choice approach. Section 9 concludes the paper and provides some elements of our future research.

## 2. Related Works

In the work reported in [11] on the Greater Paris (France), a full-day activity-plan from household travel survey is assigned to each agent. Each activity from the selected plan contains the attributes: start-time and duration. Each leg contains the transportation mode and the travel time between two successive activities. The time-choice is applied with no further changes. However, in our area of interest, using such an approach would generate an overfitting, due to the low quantity and the nature of observations in our case (which are at the department level). Indeed, we have only 590 respondents over 1,650,000 (department population) which represents only 0.03% of the total population. These observations (from the public household travel survey) are not representative of the population (only one person per household is represented).

TASHA [12] uses a bottom-up approach to generate its activity-plans, the scheduling (time-choice) process uses a predefined order of insertion (of the activities), a set of static rules apply for a set of activities. These rules concern the start-time and the duration of activities, and they are used to select the feasible proportions for both start-time and

duration of the selected activity. First, the start-time is selected, and then a duration is selected based on that start-time. Once the activity is inserted, a specific process is applied to solve the generated conflicts due the overlapping of the inserted activities. However, this approach is not suitable in our study-case, because the full chain of activity is already generated (a top-down approach). In the proposed approach, the selection of start-time (end-time) and duration takes in consideration the remaining activities and their temporal-constraints. The selection of both start-time and duration will not generate conflicts.

Zeid et al. [13] presents an approach to model time-of-day choice in a context of the tour-and activity-based models; this approach estimates the time-of-day choice based on a utility function, it chooses the arrival and departure time (from an activity) from a time-period set.

Ettema et al. [14] introduces an approach to model both the start-time and duration for an activity; it uses a utility function, which is based on the duration of activities, the preferred time to participate in activity, and the impact of schedule delay on the valuation of activities. This model is evaluated against data from the Netherlands.

Vovsha and Bradley [15] proposes a hybrid approach of the discrete choice departure time and duration models. It is employed to schedule the individual travel tours for the model MORPC; this approach uses a utility structure, which combines the advantages of both models.

In our case study, we do not use a utility function to estimate the activity time or duration. Activity-choice and their locations were previously addressed in [16], and we use MATSim (Multi-Agent Transport Simulation [17]) framework for demand-modeling and agent-based mobility-simulation: it is open-source and widely used to implement large-scale activity-based modeling.

## 3. Problem Statement and Contributions

### 3.1. Temporal Aspects of Activities in a Daily Planning

The synthetic population represents a realistic population in an artificial environment. This population is used by an activity-based model to generate an activity-plan for the different individuals describing their daily journey. A plan is composed of a set of activities in which the individual will participate and a set of legs that represent its trips between two activities. Each activity has a start-time, end-time, and a duration.

In a perfect case, a person is able to apply his daily activity-plan, by participating in all activities with a minimum duration. The simple method to choose the time of the activity and its duration is to select a start-time for the first activity, set the activity-duration, and then calculate its end-time and the travel time from the current activity to the next activity. This travel time is used to estimate the start-time of the next activity. Several constraints should be considered to represent a real-life activity time-choice: the activity-location has an open and close time (time-windows), and the activity itself has a preferred time. To deal with these constraints, capturing the temporal constraint of activities is necessary. In this work, we propose an approach that takes into consideration the previous constraints.

The objective of this approach is to set a realistic time-choice for the activity-chain by capturing the temporal constraints of future activities and by using the start-time, end-time, and duration distributions as references. These distributions are extracted from a survey data.

Survey data are first cleaned, then the different distributions are built, and the desired activity-chains are generated (or loaded). The proposed approach is applied for each activity-chain: at each step, the temporal constraints of the future activities are captured, a start-time is calculated, and a duration and an end-time value are selected respectively.

The proposed activity-chains are home-based tours, where a chain contains at least one tour. The selected activities are divided into two classes: primary activities regroups Work and Education and secondary activities regroups Leisure, Shopping, and Restaurant. Two particular activities to cite: Home and Kindergarten. Home is an mandatory activity (each activity chain has to begin and end to home); Kindergarten activity is assigned to one

of the workers-parents who has a child under 6 years. Primary activities and kindergarten always belong to the first home-tour.

An overview of the proposed model components is presented in Figures 1 and 2.

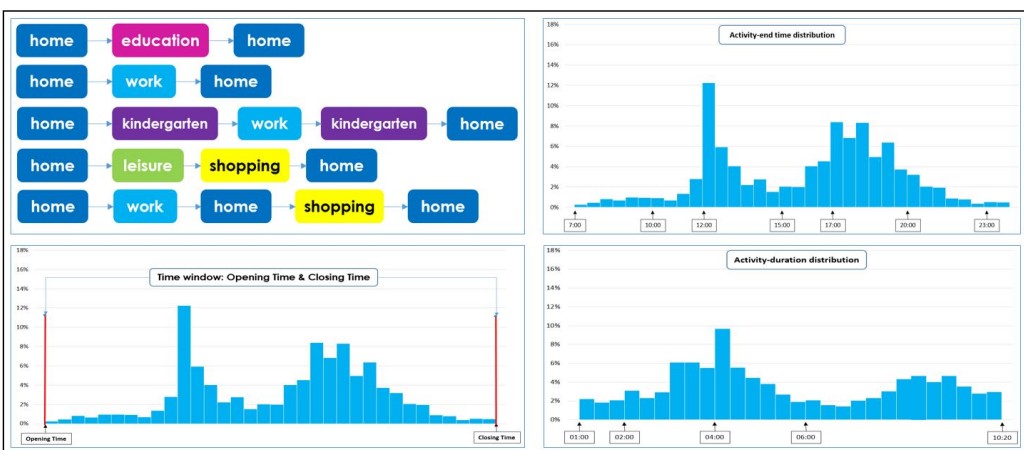

**Figure 1.** An overview of the model inputs. (**Top-left**): examples of activity-chains (home-based tours); (**bottom-left**): the activity-constraints represented by the time-window; (**top-right**): activity-end time distribution; (**bottom-right**): activity-duration distribution.

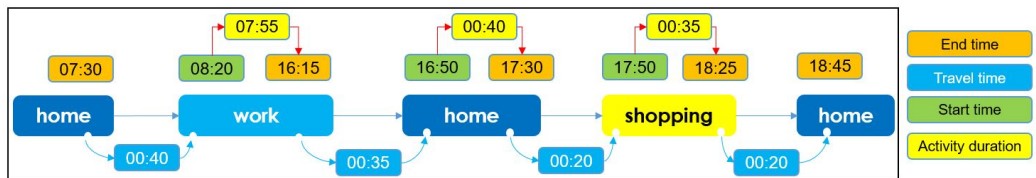

**Figure 2.** An overview of the model output: activity-settings (start-time, duration and end time) and trip-travel time.

## 3.2. Contributions and Limitations of the Work

The main contributions of the paper are as follows:

- The proposed approach is based on public empirical data of distributions related to the temporal aspect of activities; it captures the temporal constraints based on the next activities to select time-choices (duration and end-time) of the current activity. A more classical approach would use socio-economic data to determine parameters of utility-based method: these socio-economic data are usually related to individuals and requires a complex and arbitrary population segmentation process. We believe our method is more simple and straightforward.
- Other approaches that use the process of serial insertion of activity often face scheduling conflicts between two activities with overlapping time distribution: they usually proceed by removing one activity or reducing its duration. Our method has the ability to avoid this kind of coarse approximation by taking into account the following activity hours when scheduling the preceding activity.
- On other approaches based on "time budget", the duration and travel time values are independent of location: duration and travel time are computed first, then an activity-location is chosen to fit the remaining time of the "budget". Our approach is more realistic with a process that starts with the choice of activities to conduct during the day, then the location-choice for these activities based on a gravitation model, then a transportation mode-choice based on travel distance, and only then is the activity-time estimated. In this approach, one adapts the hours instead of adapting the location of its activity.
- Over the region of interest, the greater Paris, the reference method Eqasim [11] is based on much finer data that are not publicly available: the precise schedule of



activities of a subsample of the population. To generate a larger synthetic population from this subsample, the algorithm applies, for each activity, a random sampling of the departure time around the real departure time within a 30 min range. In our experience, this algorithm can lead to negative activity-duration for some agents, with a start-time higher than the end-time of the activity. In our approach, the trips departure time (activity end-time) are generated based on end-time distribution of each activity and satisfied following activities constraints.

- The proposed approach's benefit for models that use MATSim is that it improves the quality of the initial plans (in terms of score), which allows MATSim to perform the plans with a better quality; moreover, it ensures, for each agent, the feasibility of its plan (i.e., that the agent participates in all of their activities) and keeps a realistic distributions of activities for the different attributes, namely end-time and duration.
- Our model is detailed and reproducible. It also uses only public data and can therefore be directly applied to any case study on any French territory. To the best of our knowledge, this is the first detailed proposal of this kind.

The main constraints of the model are: time distributions (duration, end-time) for activities; travel speed distribution for transport modes; and demand file (the activity-chains of agents).

Some limitations and hypotheses have been made and can be listed as follows. One of the limitations of the algorithm is that it cannot extend its modeling to patterns of mobility that are not already and specifically sensed.

- The algorithm is a sequential approach that starts with generating the sequence of activities, then location-choice and mode-choice (detailed in [16]), and finally, the proposed time-choice process. At this level, the time-choice process cannot modify the activity-chain.
- The proposed model does not consider socio-economic variables (age, employee status, etc.) in the time-choice process. For example, the older (or retired) agents have a tendency to do the shopping activity early in the day. The time-choice process does not consider the traffic status, when it selects the activity-start/end-time or when it calculates the travel-time to the next activity.
- The model cannot infer activity patterns that are absent in the data, such as mobility of people who go back home for lunch or those going shopping during the lunch break for instance.
- The typical day modeled is a day of week in a pre-COVID-19 world, and the adaptation to a post-COVID-19 scenario requires time distributions that are still unknown at the time of publication. The ENTD is the only French census publicly available describing distributions of hours of activities. It is performed once every 10 years: the last one was started in 2018–2019 and is not published yet. We made the hypothesis that these data are still relevant in a pre-COVID-19 situation in terms of hours of activities, since modes of consumption and work have not changed significantly at the time of the model. In a post-COVID-19 situation times of activities, especially work hours, can be expected to change with the introduction of work-from-home and more agile model of office and therefore the time distribution of connected secondary activities.

## 4. Proposed Methodology to Set Time-Choices from Travel Survey

The objective of the approach is to adjust the activity-settings (duration, end-time) based on: typical duration, tolerance (or delay), end-time distribution, travel time, and time spent (between activities). The setting of these information in an agent planning is called Time-Choice.

An activity-based model requires generating a set of activity-plans, which refer to the daily chain of activity planned by each resident of the study-area [16]. Once the activity-plans are generated, the time-choice process is applied to adjust the time settings of the different activities of the plan.

### 4.1. Constraints in a Chain of Activities

For instance, an agent has to leave Home to go to the workplace (Work activity), then he has to come back to home (doing a home activity), go to a park-center with his children (Leisure activity), pass by a supermarket (Shopping activity), and finally return Home.

Let us consider a set of constraints related to the activities: each activity has a typical duration (the time that an agent spends in the activity-location), and several activities have a time-window (open and close time). Moreover, these activities have a preferred time (start-time) which is represented by a distribution of time (start-time or end-time), the time-preference can be deduced based on the behavior of the population (observations) who participate in a specific activity. To have a realistic time-choice, the agent has to be able to (1) participate in the all activities of his plan and (2) participate in each activity with typical duration time. For the whole population, we tend to have a distribution of start/end-time, which is correlated with the activity-preferred times (distribution), and the agent and population constraints are represented in Figure 3.

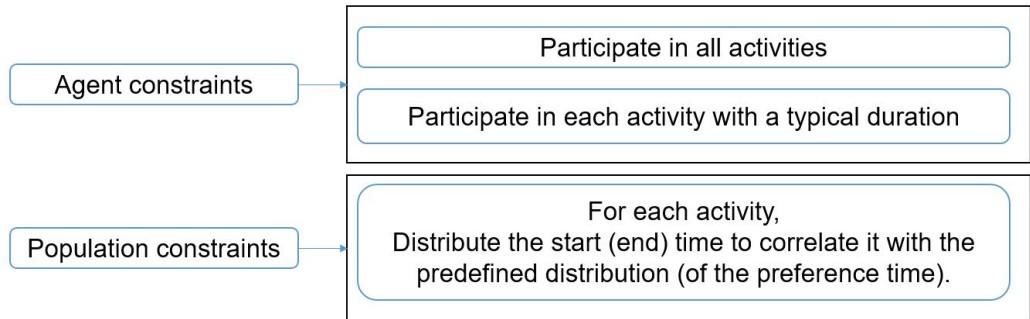

**Figure 3.** Agent-related and population-related constraints.

### 4.2. Survey Data Description

One of the main data sources that we could find is the National Transport and Travel Survey (*Enquête Nationale Transports et Déplacements*) *ENTD*, it is a French national survey with several objectives: learning about the trips of households residing in metropolitan France and their use of both public and individual modes of transport. This survey describes all trips, regardless of the reason (origin and destination of the activity), distance, duration, mode of transport used, time of year, or time of day [18].

Several types of information can be extracted from the *ENTD* trips, such as the following data distributions: (1) activity end-time, (2) activity-duration, and (3) the travel speed of each mode of transport. Others parameters can also be extracted from *ENTD*: (4) the typical activity duration, and two tolerance (delay) coefficients, namely (5) duration delay coefficient and (6) mode-travel time delay coefficient.

To apply the time-choice process, the parameters (1–3) are necessary, along with the demand file, which contains the activity-plans for the different agents; the parameters (4–6) are optional, as are (7) the activity-gap data. The required data (variables) are presented as follows:

- *ENTD* distributions:
    - Duration-distribution: each activity has a duration-distribution, which represents the proportion of users who participate in this activity for a given duration.
    - End-time distribution: for each activity, the end-time distributions can be reconstructed from *ENTD* observations, regrouped, and based on a time-period.

- – Travel speed-distribution: in this approach, each transport mode has a travel-speed value (an approximate value), it was used to estimate the travel time from an activity to another one. Four transportation modes are considered: car, public transportation, bicycle, and walking; for each of these modes, a travel speed-distribution is extracted from the observations based on the travel distance and the travel time.
- Activity-chains (demand file): activity-sequence and transport-modes of agents is the main input of this approach, these plans are loaded without the temporal attributes (duration, start-time, or end-time).
- Optional parameters:
  - – Activity gap: some activities require a gap-duration; we define the gap value as the minimum time between two activities of the same type. An example of such activity is *kindergarten*: a parent has to drop off their child to the kindergarten; the child spends at least the *kindergarten-duration* there (at the kindergarten-place), which means that the parent cannot come back to pick him up (which means a second *kindergarten*-activity) before this time (*kindergarten-gap*) is over (Figure 4).
  - – Activity-duration tolerance coefficient: the activity-duration tolerance (a tolerance refers to a positive or negative delay) represents the proportion of time that an agent can spend in an activity, more or less comparable to the typical (selected) duration. Two duration delay coefficients are defined: (1) static coefficient, which depends on the type of activity (independently from the selected duration of the activity); (2) dynamic coefficient depends on the selected duration, and is used if the duration distribution is available. These coefficients are used to control the duration delay value, with the aim to keep the duration within limits (minimum and maximum) values defined by the distribution.
  - – Travel time delay coefficient: each trip (transfer from an activity to another) has a travel time. In real life, an agent can travel faster or slower than the estimate time; this difference of time is called a delay, and the delay coefficient is used to define the maximum delay of travel time using a travel-mode .

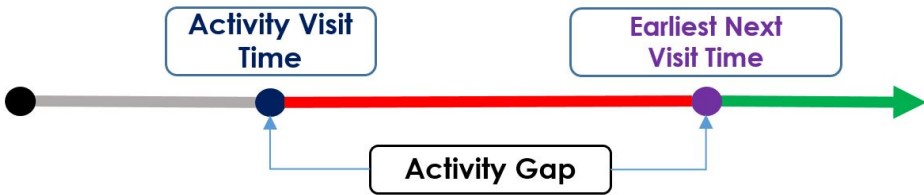

**Figure 4.** Activity gap represents the time between the two visits (the red axis), and the agent can return and do the activity only after this gap time is passed (the green axis).

*4.3. Time-Choice Setting: Overview*

This section describes the process of setting start-time, end-time, and duration for each activity of an agent planning.

For each activity of a plan, the following variables are calculated:

- Potential current activity latest time: this references the latest time possible to end (leave) the current activity. This parameter depends on three components:
  - – Maximum time is the latest simulation time, defined by the user. For example, for the generation of plan for a period of full day (starting at 00:00:00), the maximum time is 24:00:00.
  - – Activity latest end-time: this is the latest possible time for the current activity, namely the activity closing-time. This value is extracted from the activity end-time distribution.

- Minimum latest remaining activities' start-time with travel time and duration: this is the latest possible time for an agent to leave the current activity while being able to reach (and participate in) all the remaining activities (Figure 5).
- Earliest next possible visit time: this is defined for the activities with a gap value. It represents the earliest time that an individual can start (or end) the current activity; this parameter depends the last visit time (that activity) and the gap value.
- Potential activity end-time range: this range is calculated based on the current activity attributes, namely (1) start-time, (2) duration and delay duration, (3) earliest next possible visit time, and (4) the current activity latest time.
- Selected end-time value: once the end-time range is calculated, an end-time is selected from this range and the selection process is based on the activity-distribution. The process is as follows:
  - Select all possible end-time proportions from the activity-distribution which belong to the end-time range.
  - Calculate the selection (normalized) probability for each proportion.
  - Select an end-time proportion from the normalized distribution.
  - Randomly select an end-time value from the selected proportion.
- Generate the travel time to the next activity: this parameter is calculated based on three components, namely Euclidean distance between current and next activity, the used transport mode travel-speed, and the generated travel time delay.

The activity time-choice process is presented in Figure 6.

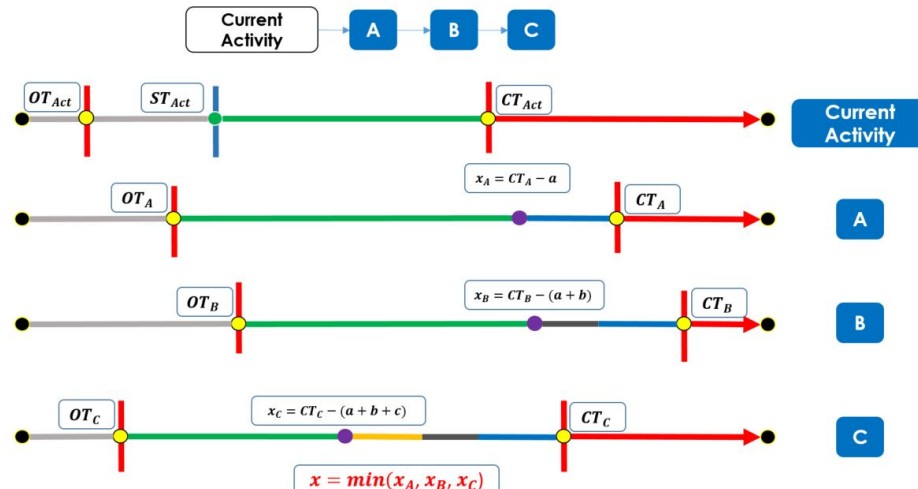

**Figure 5.** Illustration of minimum latest remaining activities start-time with travel time and duration. The following case is proposed. A user starts the current activity $Act$ at $ST_{Act}$, and he must decide the end-time of this activity based on $A$, $B$ and $C$. Each activity has a typical duration and a time-window (opening and closing time) $[OT - CT]$. The process is applied as follows: Firstly, the proposed value $x_A$ (latest remaining activities start-time with travel time and duration of the activity $A$) is calculated, the value $a$ references the sum of travel time from $Act$ to $A$ and the typical duration of $A$; $x_A$ references the subtraction of the travel time from latest start-time of $A$. Secondly, $x_B$ is calculated for the activity $B$, the value $b$ references the sum of travel time from $A$ to $B$, duration of $B$, and the calculated $a$. The same process is applied for $x_X$ of the activity $C$. Minimum latest remaining activities start-time with travel time and duration is then referenced to the minimum value $x$ from the set: $x_A$, $x_B$, and $x_C$.

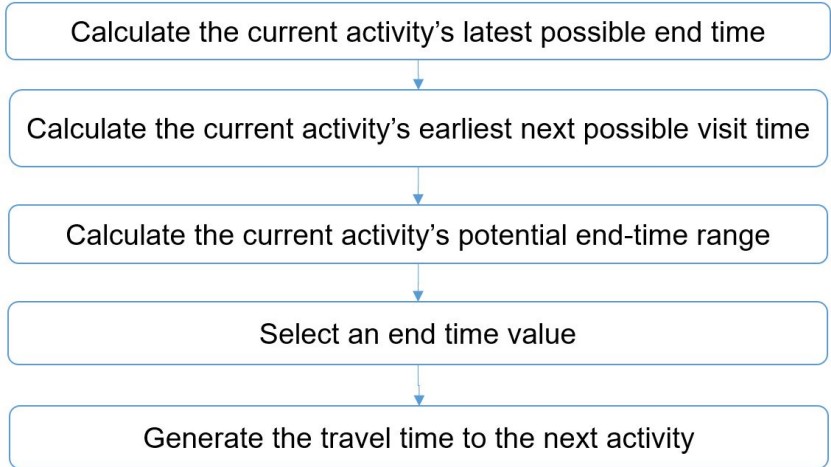

**Figure 6.** Time-choice flowchart.

## 5. Travel Demand Description

### 5.1. Trip Purpose and Transportation Mode

The ENTD data contain several tables, in this work, the focus is on *K_DEPLOC (Déplacements locaux)* table (file); this table records a set of observations (132,880 trips), and each observation (trip) has 115 attributes. Due the large number of attributes, only the attributes related to the activities and trips are selected. The activity attributes are type (trip purpose), duration, and start-time (from which end-time can be deduced). The trip attributes are transportation mode, trip distance, and travel time.

The data are split based on the nature of attributes in two groups: activities-data and trips-data. In this section, two attributes are detailed: trip purpose and transportation mode.

The trip purpose (type of activity) are divided into two categories: (a) personal activities and (b) professional activities (see Figure 7).

The personal activities are divided into eight categories (from 1 to 8). (1): Home, education and kindergarten activities; (2) shopping activities; (3) activities related to the health sector; (4) activities related to administration sector; (5) visiting a relevant or friend; (6) accompanying someone; (7) leisure activities (going to park, going to a restaurant, doing a sport, etc.); (8) going for holidays; (9) professional activities, which represent different work activities.

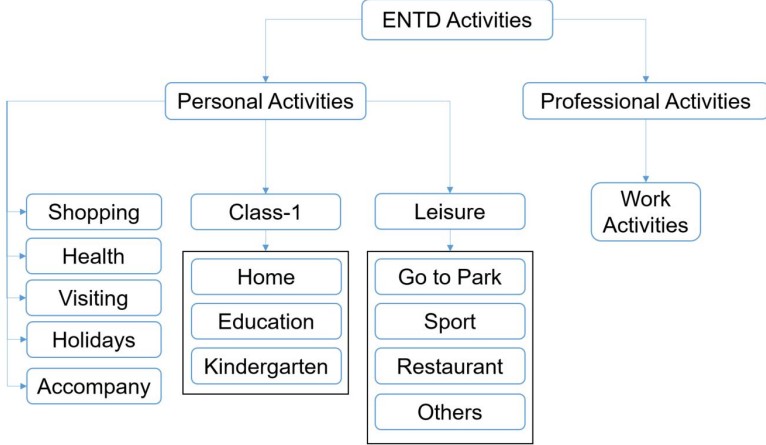

**Figure 7.** ENTD Activities—flowchart.

The selection of activities is presented as follows: *work* includes the work activity and has a fixed location; *education* contains education activities of the *class*(1) category;

*shopping* comprises shopping activities. The *ENTD*-leisure category is two-fold: *leisure* covers the activities of walking or going to parks; *restaurant*: refers to the restaurant activity. The remaining activities are not considered in the current study.

The selected modes of transport are: walking; bicycle; car (private vehicle), including both driver and passenger; and public transport, including bus, trains, and tramways.

### 5.2. Travel Demand Distributions

In this section, the travel demand distributions are detailed; they concern both duration and end-time of activities, and travel speed of transportation modes. Moreover, typical values are defined for both duration and travel speed.

### 5.2.1. Activity-Duration: Typical Value and Distribution

Two duration parameters are required: typical duration and duration distribution. The activity typical duration can be defined for each activity. In this work, the five-number summary is computed to get more information about the available data (activities data). Due the large variety of data, a first constraint is applied to restrict the duration range (minimum and maximum values). Second, the five-number summary are calculated; as result, the median value seems like the more realistic duration for an activity and is considered as the typical value. These typical durations are stored in a module called duration, which is a part of the population generator (Figure 8).

```xml
<module name="duration">
    <param name="work"        value="16800.0" />
    <param name="education"   value="14700.0" />
    <param name="leisure"     value="3420.0"  />
    <param name="shopping"    value="2040.0"  />
    <param name="restaurant"  value="4800.0"  />
</module>
```

**Figure 8.** Population generator module—typical activity-durations (in seconds).

Activity-duration distribution is built from the updated survey (ENTD) data for each activity. The observations are split based on the duration into bins with a specific range-length which depends on activity data, and the range length should be enough to avoid the bins with few observations. The different distributions are shown in Figure 9.

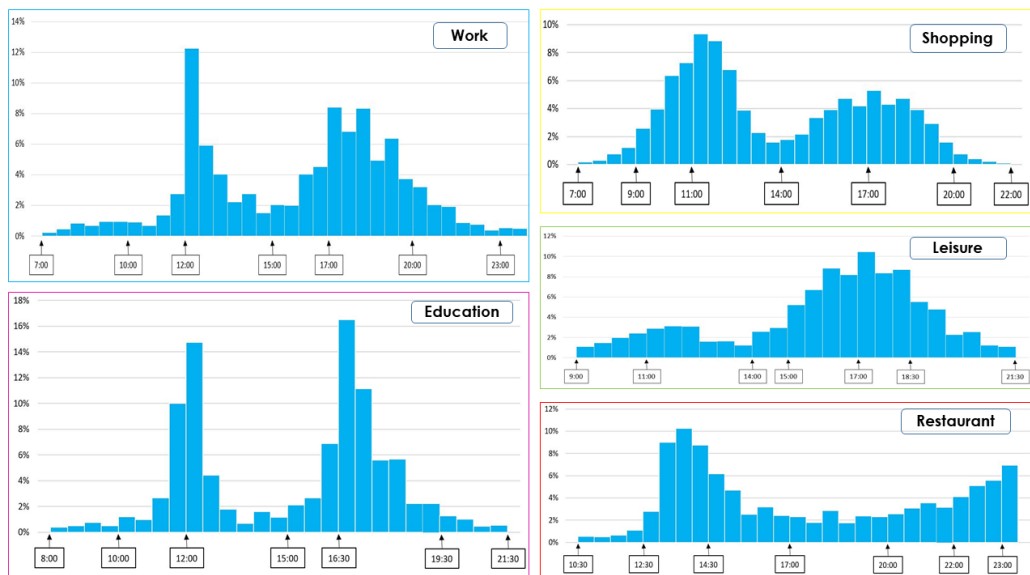

**Figure 9.** The duration distribution for the different activities.

*Home* and *kindergarten* are two particular activities. *Home* does not have a defined duration because it is assumed that an individual does not have a limited duration of staying at home. *kindergarten* has a null duration because the agent (parent) joins only the activity-place without spending time there.

### 5.2.2. End-Time Distribution

The activities' data are used to generate the end-time distribution of the different activities. These data contain the information about: type of activity, duration, start-time, and end-time. Next, as an option, a time-window is defined for each activity, which represents the realistic time-window when this activity often takes place (realistic time-period for the activity). Once the time-window is defined for each activity, the observations are divided into bins (distributions) with an equal range of time.

Due the lack of data for the kindergarten activity, the distribution is considered as uniform, from 06:30 to 19:30, with 30 min as time-range.

### 5.2.3. Travel Speed Range by Mode

The travel speed is used to calculate the real travel time between two activities. The trips-data (extracted from ENTD) contain information about the distance and travel time for each observation (trip), and the trip speed for each observation can be deduced. To generate a distribution of travel speed for each mode, the following process is applied: first, the travel speed is calculated for each trip; then, a data cleaning process is applied to remove the overlays (non defined or non realistic) data, mainly due the null speed (defined or null travel-distance). In next step, a minimum and a maximum speed value (bounds) are defined for each mode, with the aim to restrict the study data. These bounds are used to eliminate the non-realistic observations from the dataset. Finally, the valid observations are split into bins (see Figure 10).

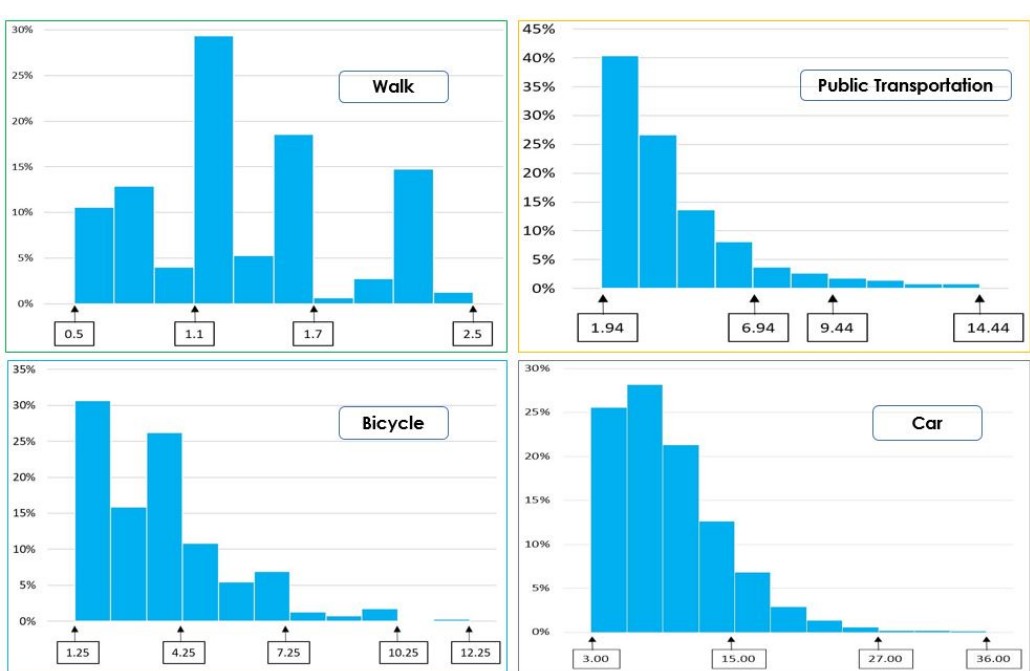

**Figure 10.** The travel speed distribution for the different modes.

As in the case of the activity-duration, a typical travel speed is defined for each mode; this typical value is used to estimate the travel time between two activities. To understand better the distribution of the travel speed, the five-number summaries are calculated for each mode. For the walk mode, the median speed is considered as typical travel speed (with 1.14 m/s). For the remaining modes, bicycle, car, and public transportation, the third

quartile $Q_3$ is more realistic (with 4.44 m/s, 11.90 m/s, and 5.00 m/s respectively) and so it considered as the typical travel speed for these modes.

### 5.3. Travel Demand Parameters

#### 5.3.1. Chains of Activity from the Demand File

After generating the different approach components—the activity and travel-speed distributions—the next step is to load the demand file. This file contains a set of plans (at least one plan) for each agent. A plan is composed of three components: (a) *typeAct*, which represents the sequence of activities; (b) *act*, which references the location of activities; and (c) *modes*, which represents the sequence of the used transport-modes (see Figures 11 and 12).

```
<agent id="agent_1" >

  <plan    typeAct ="home#work#home#leisure#home"      ◄─── Activities-types
           act     ="h2549#7clk_#h2549#l32#h2549"        ◄─── Activities locations
           modes   ="car#car#walk#walk"  ◄───────────────   Transport modes
  />

  <plan    typeAct = "home#work#home#leisure#home"
           act     = "h2549#7clk_#h2549#LaVallee_l4#h2549"
           modes   = "car#car#walk#walk"
  />

</agent>
```

**Figure 11.** Agent activity-plans: *agent*_1 has two plans, both of which have the same sequence of activities, the sequence of used transport modes, *home* and *work* locations; however, they differ in the location of *leisure* (*l*32 for the first plan and *LaVallee_l*4 for the second plan).

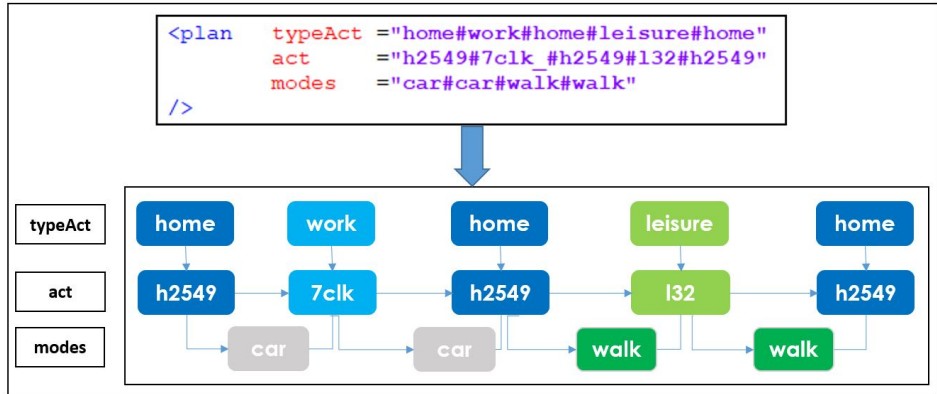

**Figure 12.** Activity-plan: the first plan of the *agent*_1 is detailed, the sequence of activities is *home* → *work* → *home* → *leisure* → *home*. The locations of activities are: *h*2549(*home*), 7*clk*(*work*) and *l*23(*leisure*). The transport modes are: *car* for trips *home* ↔ *work* and *walk* for trips *home* ↔ *leisure*.

After generating the different main data, the next step is to load the optional data, namely the activity-gap and the delay coefficient for both activity-duration and travel time.

#### 5.3.2. Activity-Gap for Kindergarten

As cited previously, some activities require a gap-value. In this study, only the *kindergarten* activity is considered by this value; due the limited information about this activity (in *ENTD*), $Gap_{kindergarten}$ is equal to four hours. To integrate this components into the synthetic generator, a defined module *gap* is created, it contains for each entry (activity), its gap-value.

5.3.3. Tolerance on Activity-Duration and Travel Time

In this section, we focus on the static tolerance coefficients for both duration and travel time. These parameters are defined independently from the survey data, also as the selected duration and travel speed (which is used to calculate the travel time). Tolerance represents the proportion of time (freedom) that is given to a traveler: to spend more or less time in an activity (the duration delay) or to go faster or slower (travel time delay). The initial static coefficients of duration vary from 0 for work to 20 percent for restaurant. In our experiment, coefficient $TTDCoef$ is arbitrarily set to 20% for each mode.

To explain well the meaning of these coefficients, we consider the case of work-activity; the typical duration is set to (4 h 45 min = 285 min), the tolerance coefficient equals to 10%, which means that the tolerated duration is equal to 28.5 min.

For both home and kindergarten activities, the duration delay set to 0. For the remaining activities, the tolerance coefficient has a converse relationship with the duration of activity. The travel tolerance coefficient is set to the same value for all transport modes.

Two tolerance (delay) modules are then defined: activity-delay and mode-delay.

**6. Algorithm for Time-Choices Setting**

In this section, the time-choice process is explained in details. In the first part, a set of variables is defined. Due the large number of variables, only the general ones are explained; the remaining variables are explained further along om the process. In the second part, a set of components is calculated, which are used to capture the temporal constraints of the current activity. Finally, these components are used to select the duration and end-time of the current activity.

The list of variables is presented as follows:

- $x, y$: two activities from the activity-plan.
- $A$: current activity.
- $B$: next immediate activity to $A$ in the activity-plan.
- $N_A$: remaining activities (that follow $A$ in the activity-plan).
- $z$: remaining activity $z \in N_A$.
- $ET_A$: earliest end-time of $A$.
- $LT_A$: latest end-time of $A$.
- $EST$: earliest (simulation) time (in this study, it is defined as 05:30).
- $duration_x$: typical duration of $x$.
- $Gap_x$: gap value of $x$.
- $DDist_x$: duration distribution of $x$.
- $D_x$: end-time distribution of $x$.
- $e_x$: end-time value from the distribution $D_x$ ($e_x \in D_x$).
- $e_x^*$: latest end-time value from the distribution $D_x$.
- $Ee_x^*$: earliest end-time value from the distribution $D_x$.
- $d_{x \to y}$: euclidean distance from $x$ to $y$.
- $mode_{x \to y}$: transport mode used to go from $x$ to $y$.
- $travelSpeed_m$: travel speed of the transport mode $m$.
- $C_{x \to y}$: activity-set from $x$ to $y$.
- $|C|$: cardinal of $C$.
- $C_{First}(C_0)$: first element of $C$, $C_{First} = x$.
- $C_{Last}(C_{|C|-1})$: last element of $C$, $C_{Last} = y$.
- $DDCoef_x$: duration delay coefficient of $x$.
- $DyDCoef_x$: dynamic duration delay coefficient of $x$.
- $StDCoef_x$: static duration delay coefficient of $x$.
- $TTDelay_{x \to y}$: travel time delay from activity $x$ to activity $y$.
- $TTDCoef_m$: transport mode $m$ travel time delay coefficient.

- $P$: proportion-time set.
- $p$: proportion-time, $p \in P$.
- $r_p$: distribution rate of $p$.
- $S_p$: selection probability of $p$.

### 6.1. End-Time Range Components

The defined end-time range has several components: current (local) activity latest time ($LLT$) and earliest next visit time ($EVT$) are used to define the bounds of the end-time range based on the remaining activities (for $LLT$) and previous activities (for $EVT$). A more specific range is calculated from start-time and a selected duration for the current activity, a combination between these two ranges defines the final end-time range.

#### 6.1.1. Local Latest Time ($LLT$)

$LLT_A$ represents the local latest time for the current activity $A$; it depends on the following components:

- $LST$: this represents the latest (simulation) time, in this study, defined as 24:00:00.
- $e_A^*$: this is the last value of end-time distribution $D_A$ (see Equation (1)).
- $ST\_TTD_A$: this represents the latest possible time in which reaching and execution all the following activities are guaranteed (see Equation (2)).

$$e_A^* = \max_{e_A \in D_A} (e_A) \tag{1}$$

From a mathematical point of view, $ST\_TTD_A$ is equal to the minimum value of $ST\_TTD_{A \to z}$, where $z \in N_A$ (see Equation (2)).

$$ST\_TTD_A = \min_{z \in N} (ST\_TT_{A \to z}) \tag{2}$$

To conclude, $LLT_A$ is equal to the minimum of: $LST$, $e_A^*$ and $ST\_TTD_A$ (see Equation (3)).

$$LLT_A = \min(LST, e_A^*, ST\_TTD_A) \tag{3}$$

The description of each $LLT$ components is presented in Figure 13.

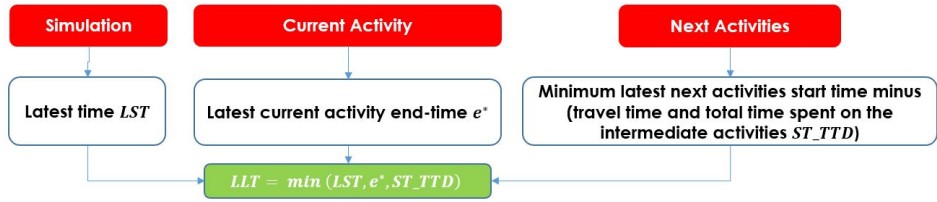

**Figure 13.** Local latest time: $LLT$ is equals the minimum of (1) $LST$ from simulation, (2) $e^*$ of the current activity, and (3) $ST\_TTD$ the latest possible time to complete the activity-plan (next activities), by satisfying the duration and end-time constraints (spending the typical duration in each activity and finishing each activity before the closing-time).

#### 6.1.2. Start Time Minus Total Travel Time and Durations $ST\_TTD_{A \to z}$

In this section, the calculation of $ST\_TTD$ is detailed. $ST\_TTD_{x \to y}$ represents the difference between (1) the total travel time from $x$ to $y$ and the total of duration of all intermediate activities $I_{x \to y}$ and (2) the latest possible start-time of activity $y$.

First, the start-time $ST_x$ of the activity $x$ is defined as equal to the difference between (1) the typical $duration_x$ and (2) the end-time $e_x$ (see Equation (4)).

$$ST_x = e_x - duration_x \tag{4}$$

Then, the latest possible start-time $lpST_x$ of activity $x$ is calculated. It refers to the difference between $duration_x$ and $e_x^*$ (see Equation (5)).

$$lpST_x = e_x^* - duration_x \tag{5}$$

$I_{x \to y}$ represents the set of activities visited (intermediates) along on the path from $x$ to $y$. It refers to all activities from $x$ to $y$ exclude $x$ and $y$ (see Equation (6)).

$$I_{x \to y} = C_{x \to y} - \{x, y\} \tag{6}$$

$pTT_{A-z}$ represents the predicted travel time of the whole path from $A$ to $z$, passing by all the intermediate activities ($I_{A \to z}$) (see Equation (7)).

$$pTT_{A-z} = \sum_{i=0}^{|C|-1} pTT_{C_i \to C_{+1}} \tag{7}$$

$pTT_{x \to y}$ is the predicted travel time between $x$ and $y$. It is calculated based on the Euclidean distance $d_{x \to y}$ and the travel-speed from $x$ to $y$ (see Equation (8)).

$$pTT_{x \to y} = \frac{d_{x \to y}}{travelSpeed_{mode_{x \to y}}} \tag{8}$$

$TD_I$ is the total typical duration of all activities that belong to the intermediate set $I$ (see Equation (9)).

$$TD_{I_{A \to z}} = \sum_{y \in I_{A \to z}} duration_y \tag{9}$$

Finally, $ST\_TTD_{A \to z}$ can be defined as in Equation (10).

$$ST\_TTD_{A \to z} = lpST_z - (pTT_{A \to z} + TD_{I_{A \to z}}) \tag{10}$$

The different calculated parameters are detailed in Figures 14 and 15.

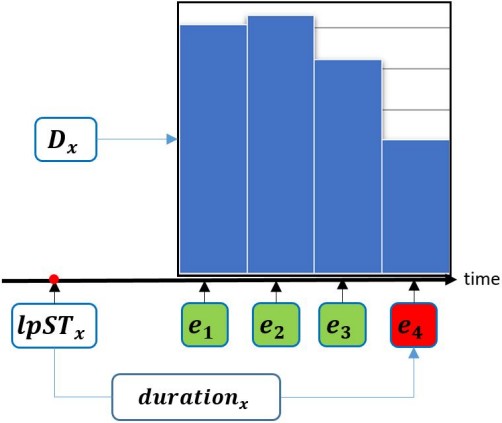

**Figure 14.** Latest possible start-time for next an activity $x$.

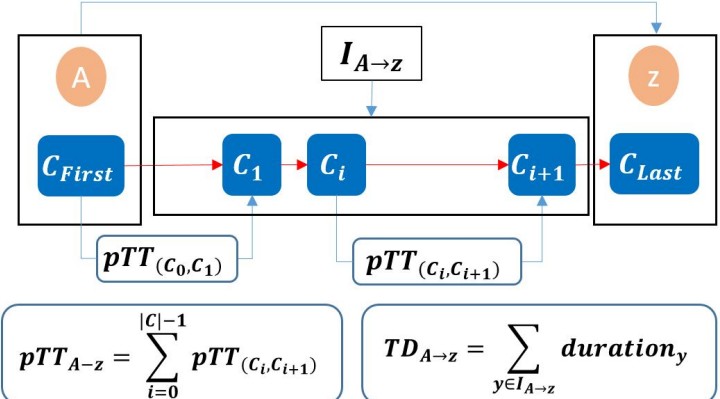

**Figure 15.** Predicted travel time and time spent on the intermediate activities: the sum of the travel time from $A$ to $z$ is the predicted travel time $pTT_{A \to z}$, and the sum of duration of the intermediate activities from $A$ to $z$ is the time spent $TD_{A \to z}$.

6.1.3. Potential Activity End-Time Range

After calculating the current (local) latest time $LLT_A$, the next step is to calculate the end-time range, this range has two potential bounds: (a) lower bound, which references the earliest end-time $ET_A$, and (b) upper bound, which references the latest end-time $LT_A$. To generate this range, the following steps are applied.

First, the start-time $ST_A$ is initialized: if $A$ is the first activity, then start-time of $A$ is equal to $EST$. The second step is to select a duration $dur_A$ for the current activity $A$ (from the duration distribution $DDist_A$ of $A$). The selection process is as follows: a duration proportion is selected, and then a random duration (from this proportion) is selected. If the duration distribution is not available, $duration_A$ is set to $dur_A$. After calculating $ST_A$ and $dur_A$, the initial end-time $IET_A$ is deduced (see Equation (11)).

$$IET_A = ST_A + dur_A \tag{11}$$

The next step is to calculate the duration delay. To do so, the dynamic duration delay coefficient $DyDCoef$ is calculated. This parameter is used to keep the duration of activity within the duration distribution limits. To do so, $DyDCoef$ must satisfy the following constraints: it must have a positive value (Equation (12)), and the selected duration, including its delay has to be greater than the minimum duration $minDur_A$ and less than the maximum duration $maxDur_A$ (see Equations (13) and (14)).

$$DyDCoef \geq 0 \tag{12}$$

$$DyDCoef_A \leq \frac{maxDur_A - dur_A}{dur_A} \tag{13}$$

$$DyDCoef_A \leq \frac{dur_A - minDur_A}{dur_A} \tag{14}$$

$DyDCoef_A$'s value is drawn from the available range.

If the static delay coefficient (which depends on the type of the activity) is included, the duration delay $DDCoef_A$ can be deduced, which equals to the minimum value of both delay coefficients (see Equation (15)).

$$DDCoef_A = \min(StDCoef_A, DyDCoef_A) \tag{15}$$

Once the $DDCoef_A$ is defined, the duration delay $DD_A$ can be calculated (see Equation (16)).

$$DD_A = DDCoef_A \times dur_A \tag{16}$$

The end-time range bounds depend on: earliest end-time $Ee_A^*$, initial end-time value $IET_A$ and delay duration $DD_A$. The range bounds can be deduced as in Equations (17) and (18).

$$ET_A = max(Ee_A^*, IET_A - DD_A) \tag{17}$$

$$LT_A = max(Ee_A^*, IET_A + DD_A) \tag{18}$$

Figures 16 and 17 illustrate this process.

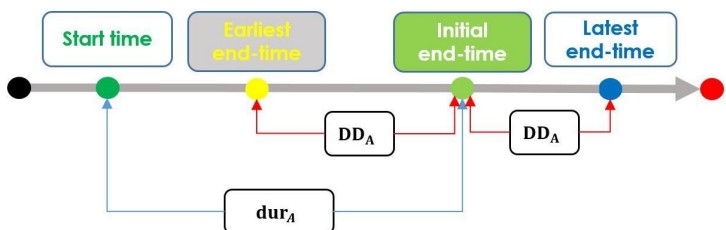

**Figure 16.** End-time—detailed view: from the start-time *ST* and the selected duration *dur*, an initial end-time value is calculated; when the tolerance coefficient is included, a tolerance (delay) duration *DD* is then estimated, which combines with the initial end-time value to define the end-time range.

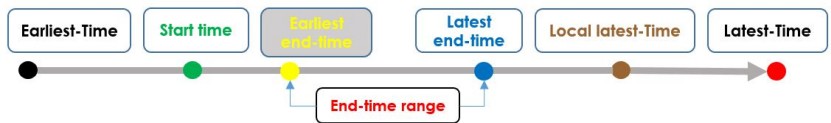

**Figure 17.** End-time range-global view: both end-time range bounds are located between the start-time and the latest-time.

### 6.1.4. Earliest Next Visit Time

After the different parameters of the activity $A$—$LLT_A$, $ET_A$, $LT_A$—are calculated, the next component is the earliest next visit time $EVT_A$. It is only necessary if the activity $A$ satisfies the following constraints: it has a gap-value ($Gap_A > 00{:}00{:}00$) and it was visited already, which means that the last visit time parameter $LVT_A$ of the activity $A$ is not null ($LVT_A <> null$). $EVT_A$ is calculated following Equation (19).

$$EVT_A = LVT_A + Gap_A \tag{19}$$

Both last visit time *LVT* and earliest next visit time *EVT* are represented in Figure 18.

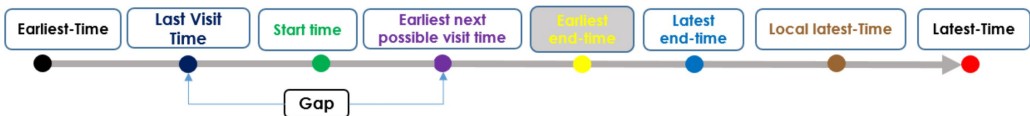

**Figure 18.** Earliest next visit time: it is located between the start-time and the latest-time.

### 6.1.5. Update the End-Time Range

After $LLT_A$ (see Equation (3)), the initial end-time range bounds $ET_A$ and $LT_A$ (see Equations (17) and (18)), and the earliest next visit time $EVT_A$ (see Equation (19)) are calculated, the next step is the generation (updating) of the final end-time range. The lower bound is updated as the maximum value between $EVT_A$ and $ET_A$ (see Equation (20)), while the upper bound is updated as the minimum value between $LLT_A$ and $LT_A$ (see Equation (21)).

$$ET_A = \max(EVT_A, ET_A) \tag{20}$$

$$LT_A = \min(LT_A, LLT_A) \tag{21}$$

The updated end-time range is represented in Figure 19.

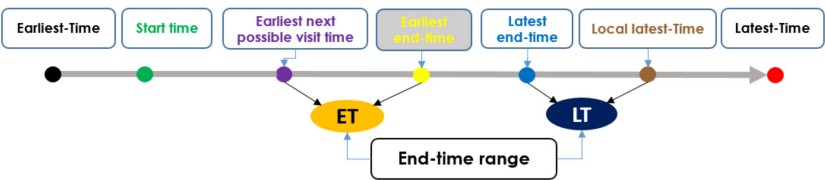

**Figure 19.** Updated end-time range: the final end-time range is generated, which includes the constraints $EVT$ and $LLT$ on the initial end-time range; therefore, the range-bounds are updated.

*6.2. Activity-End-Time: A Final Value*

6.2.1. Select an End-Time Value

After the end-time potential range is calculated, the next step is to to select an end-time value from that range. Based on the nature of the activity, two main cases are considered: home activity case and other activities case.

The activity *home* does not have a defined duration; departure time (home end-time) depends on the beginning (start-time) of the next activity $B$. To calculate it, the home activity end-time distribution is updated as the next activity start-time distribution (see Equation (22)). For the remaining activities, the end-time distribution is kept the same (no updating process).

$$D_{home} = ST_B \tag{22}$$

After updating the activity end-time, the selection process of an end-time value is applied.

First, all possible end-time proportions $p \in P$ are selected, which satisfies the following constraints: it is a proportion from $D_A$, and the intersection between $p$ and the end-time range is not empty (see Equation (23)).

$$(p \in D_A) \wedge (p \cap [ET_A, LT_A] \neq \varnothing) \tag{23}$$

Second, the selected proportions $P$ are normalized. Due the intersection of the end-time range and $P$, some proportions partially belong to the end-time range. An adjustment process is necessary to adapt the weight $r$ of these partially proportions. This process considers the quantity of time from the proportion that belongs to end-time range. A belonging parameter $b_p$ is defined for a proportion $p$ as the ratio of $p$, which belongs to $[ET_A, LT_A]$ (see Equation (24)).

$$b_p = \frac{|p \cap [ET_A, LT_A]|}{|p|} \tag{24}$$

The proportion weight is adjusted following Equation (25), and it is illustrated in Figure 20.

$$r_p = b_p \times r_p \tag{25}$$

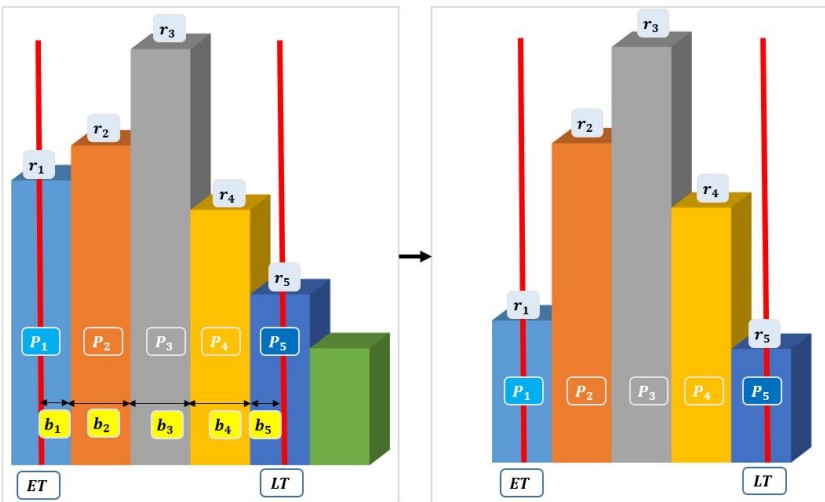

**Figure 20.** Adjustment process for proportions weight (**left**): the proportions belong (totally or partially) to the end-time range are selected, and its belonging rate is calculated; (**right**): the selected proportions' weights are adjusted based on the belonging rate.

Once the adjustment is done, a selection probability $S_p$ is calculated for each proportion $p \in P$ (see Equation (26)).

$$S_p = \frac{r_p}{\sum_{x \in P} r_x} \qquad (26)$$

Third, an end-time proportion $p^*$ is selected from $P$ based on $S$. Finally, a random value $ed$ is selected from the selected proportion $p^*$. $ed$ has to satisfy the constraints of Equation (27).

$$ed \in p^* \wedge ed \in [ET, LT] \qquad (27)$$

The process of selecting an end-time value is presented in Figure 21.

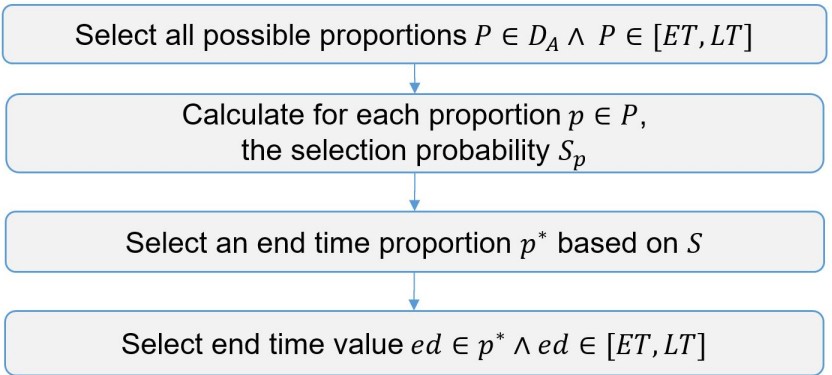

**Figure 21.** Select end-time value process.

### 6.2.2. Update the Activity-Attributes

After an end-time value is selected, the next step is to update the main activity attributes for the current activity $A$: activity-duration (see Equation (28)) and end-time (see Equation (29)).

$$A_{max\_dur} = ed - ST_A \qquad (28)$$
$$A_{end\_time} = ed \qquad (29)$$

The end-time for home activity is equal to $ed$ minus the travel time $TT_{home \to B}$ from *home* to the next activity $B$ (see Equation (30)).

$$home_{end\_time} = ed - TT_{home \to B} \qquad (30)$$

*6.3. Next Activity: Travel and Start-Time*

6.3.1. Travel Time to the Next Activity

In this step, the travel time $TT_{A \to B}$ from $A$ to the next activity $B$ is estimated. To calculate this travel time, two measures are necessary: the predicted travel time $pTT_{A \to B}$ and the travel time tolerance (delay) coefficient $TTDelay_{A \to B}$.

The calculation of $pTT_{A \to B}$ requires the following: the Euclidean distance $d_{A \to B}$ and the selection of a travel speed $travelSpeed_{mode_{A \to B}}$ (see Equation (8)). In contrast to $pTT_{x-z}$ (predicted travel time of the path from $x$ to $z$ via a set of activities $I_{x \to z}$), which uses the typical travel speed as reference, $pTT_{A \to B}$ uses the travel speed distribution to select the speed value.

As for the activity-duration, a travel speed of the used transport $mode_{A \to B}$ is selected from the distribution. If this distribution is not available, the typical value is then used. The coefficient $TTDCoef_{mode_{A \to B}}$ is used to calculate the travel time tolerance (delay) $TTDelay_{A \to B}$ (see Equation (31)).

$$TTDelay_{A \to B} = TTDCoef_{mode_{A \to B}} \times pTT_{A \to B} \tag{31}$$

From the calculated $TTDelay_{A \to B}$, two values are estimated: the quickest travel time $QTT_{A \to B}$ and the slowest travel time $STT_{x \to y}$. These values represent the defined travel time range (see Equations (32) and (33)).

$$QTT = pTT_{A \to B} - TTDelay_{A \to B} \tag{32}$$

$$STT = pTT_{A \to B} + TTDelay_{A \to B} \tag{33}$$

The travel time $TT_{A \to B}$ is then randomly selected from the range $[QTT, STT]$ (see Equation (34)).

$$TT_{A \to B} \in [QTT_{A \to B}, STT_{A \to B}] \tag{34}$$

6.3.2. Start-Time of the Next Activity

The start-time $ST_B$ of the next activity $B$, is equal to the sum of the end-time $A_{end\_time}$ and the travel time $TT_{A \to B}$ from $A$ to $B$ (see Equation (35)).

$$ST_B = A_{end\_time} + TT_{A \to B} \tag{35}$$

In the case of an activity $B$, concerning the gap-value $Gap_B <> null$, the earliest next visit time $EVT_B$ is checked (see Equation (19)). To avoid waiting for the next activity, the agent has to start his next activity at $EVT_B$ minus $duration_B$; to calculate the earlier start-time, the $duration_B$ is subtracted from $EVT_B$ (see Equation (36)).

$$ST_B = EVT_B - duration_B \tag{36}$$

As a consequence, two potential start-time $ST_B$ values are defined: $ST_B^1$ refers to Equation (35) and $ST_B^2$ refers to Equation (36). A combination of Equations (35) and (36) is presented in Equation (37)

$$ST_B = max(ST_B^1, ST_B^2) \tag{37}$$

*6.4. Time-Choices Illustration*

In this section, the process is detailed. First the activity-chain and the mode-chain are detailed, and the calculated distance and travel time of the trips data are presented. The process is illustrated on the first activities: home, kindergarten, and work.

Activity-Plan and Trips Travel Time

The selected example of the illustration is presented as follows. The activity-chain is composed of two home tours: the first tour is a sequence of trips, from home to kindergarten, then to work; after finishing the work activity, the agent has to pass by the kindergarten before arriving at home. On the second tour, the agent does a leisure activity, then

goes to shop before come back home. In the first tour, the agent uses the car as a transport mode; in the second home-tour, he uses the public transportation mode to go from home to leisure place, then walks to the shop-place and then takes public transportation again to come back home. The activity-chain and mode-chain are presented in Figure 22.

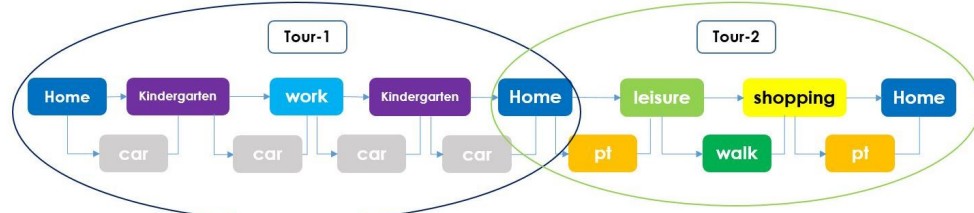

**Figure 22.** Activity plan (HKWKHLShH): the first tour represents a return trip between home and work via the kindergarten using the car as a mode; the second tour represents a home-tour passing by leisure and shopping places, and the used modes are a combination between pt and walk.

To avoid the conflict between the activities from the same type (*home*, *kindergarten*) during the illustration, these activities are renamed (see Figure 23).

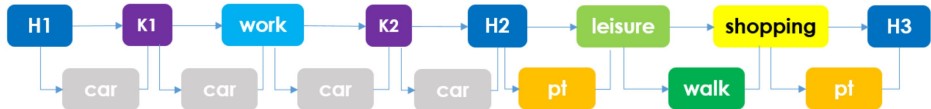

**Figure 23.** Activity plan (HKWKHLShH)—Instantiation.

The trip distance and predicted travel time are important inputs for the proposed approach. Based on the activity-location, the Euclidean distance of trips is calculated; then, the typical travel speed of the trip-mode is used to calculate the predicted travel time. The calculated distance is presented in Figure 24.

Based on the calculated predicted travel time for the different trips of Figure 24, the travel time from an activity to its next activities can be estimated. In Appendix A, we detail computations and time values returned by the algorithm for the first steps:

- Travel time from an activity to its next activities, from the initial predicted travel time;
- Estimation of the first home departure time: H1 end-time;
- Calculation of latest possible time for H1 activity: local latest time $LLT_{H1}$;
- An initial range for H1-end-time considered alone;
- Modification of H1-end-time range taking into account all the activities to be done during the day;
- Building H1 departure time distribution;
- Drawing final H1 departure time from distribution;
- Computing H1-K1 travel time and K1-start-time knowing H1-departure time;
- Estimating K1-end-time and travel time to work; no duration since it is kindergarten activity;
- Estimating work-related time-choices: duration, end-time, and travel time to next activity.

These steps are related to activities H1, K2, and *work*. The remaining activities of the daily planning are modeled the same way as *work*.

Table A1 gives a summary of the important computed times for this daily planning.

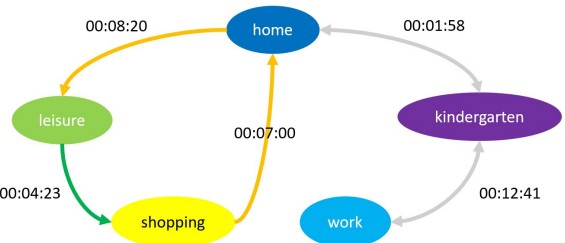

**Figure 24.** Trips' predicted travel time: the green arrow is a walking trip, the gray arrows are car trips, and the orange arrow is a public transportation trip.

## 7. Urban Experiment

This work is the continuation of the work started in [16], where we modeled travel behaviors of the population of the catchment area of a rather large future district (real estate development *LaVallée*, due in 2024). This area of study is shown in Figure 25 along with home locations. In this section, the activity-plans for the whole population of the catchment area (126,151 agents) are generated. The work experiment is divided into two parts: validation and simulation. In the first step, the generated distributions (which are extracted from the activity-plans) are compared to the base $ENTD$ distribution. In the second step, two generated activity-plans are simulated; the first one uses the proposed approach constraints, while the second does not use those constraints. The objective of the simulation step is to compare the score of both plans and to deduce the benefit of applying the proposed approach.

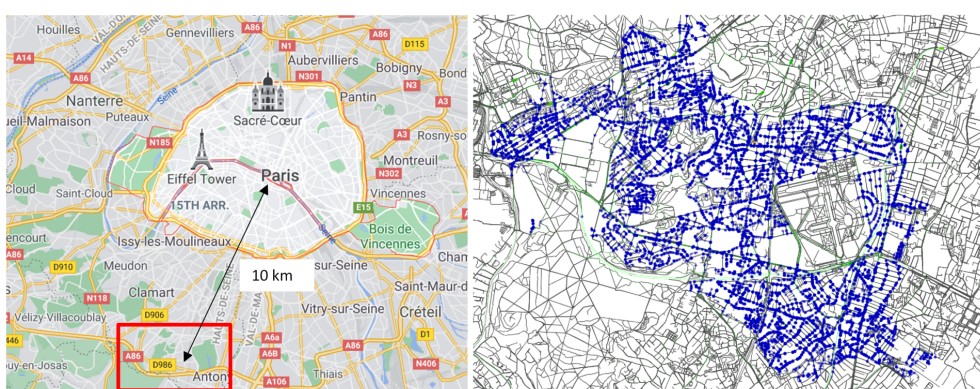

**Figure 25.** The region of interest is a suburban area 10 km from Paris, France. Its population amounts to 126k agents over five municipalities. Blue dots are Home-locations of the agents (cartography MapData ©2021 Google).

### 7.1. Model Validation

The validation model compares the distribution of the generated activity-plans, for both duration and end-time attributes, with the distributions extracted from $ENTD$. It is composed of four steps: (1) the configurations are defined based on the approach parameters, (2) the population-plans are generated for each configuration, (3) the configurations outputs are compared with $ENTD$ distributions, and (4) the best configurations with its population are selected for the simulation. The validation is applied to a set of activities $X$, which contains: work, education, leisure, shopping, and restaurant.

Firstly, if the configuration $c$ (the tolerance coefficients) is varied, a set of activity-plans are generated; they are compared to the referent distributions. Six configurations {A, B, C, D, E, F} are proposed to validate the proposed approach. The configurations (A) and (B) use only static coefficients, (A) uses the standard coefficients defined in Section 5.3, (B) and uses a double value of configuration (A). With the aim of studying the impact of static coefficients on the activity-plan times, the objective of these configurations is to test the output distribution independently of the selected duration. To avoid selecting

a non-feasible duration, the maximum value of dynamic coefficient is used to keep the selected duration within the distribution range. The configurations (C) and (E) use only the dynamic coefficients. (C) uses the maximum dynamic coefficient value and generates the largest end-time (duration) range with the selected duration as a center, while (E) uses a random dynamic coefficient. The configuration (D) does not use the tolerance coefficients; as a result, the selected duration is used immediately to calculate end-time value. The last configuration (F) is a combination of (A) and (C) and references the defined configuration are presented in the process described in Section 6.1, and the different configurations are presented in Table 1.

**Table 1.** Model validation: set of configurations.

| Conf | Static | Dynamic | Description |
|:---:|:---:|:---:|:---:|
| A | x | | Standard static coefficients |
| B | x | | Double standard static coefficients |
| C | | x | Maximum dynamic coefficients |
| D | | | Null static coefficients |
| E | | x | Random dynamic coefficients |
| F | x | x | Standard static coefficients Random dynamic coefficients |

Secondly, by using the configuration settings (tolerance parameters), the next step is to generate the activity-plans for the whole population. Once the activity-plans are generated, by basing on the $ENTD$ proportions (time-ranges), the end-time and duration distributions are extracted for each activity from the set $X$. This step is applied for each configuration separately.

Thirdly, to evaluate the different configurations $c$, a validation error measure $error_c$ is defined for both duration and end-time. For each activity $x$ from $X$, an activity-error gap $error_c^x$ (not to be confused with the activity-gap $Gap$) is calculated between the configuration $c$ output distributions and the reference distribution $ENTD$, which refers to the difference between each time-proportion $p$ and $P_x$ ($P$ refers to both duration and end-time distributions) between the reference rate $r_p^x$ and the configuration rate $c_p^x$ (see Equation (38)). Finally, the $error_c$ is equal to the average of all $error_c^x$ (see Equation (39)).

$$error_c^x = \frac{1}{|P_x|} \times \sum_{p \in P_x} |c_p^x - r_p^x| \tag{38}$$

$$error_c = \frac{1}{|X|} \times \sum_{x \in X} error_c^x \tag{39}$$

Finally, after the duration and end-time average error for each configuration are calculated, the next step is to select the best configuration. By comparing the average error of both duration and end-time error of each configuration, with a priority to the end-time. The best configuration is saved also as its activity-plans, they will be used in the simulation step.

### 7.2. Model Simulation

After the best configuration $c$ from the validation step is selected, two sets of activity-plans (for the whole population) are generated. The first set is generated using the proposed approach, which includes the defined time constraints, while the second set of plans is built without the constraints $EVT$ and $LLT$.

To establish a fair comparison of time-constraint plan generation with simulation versus no-time-constraint plan generation with simulation, five different generated plans are used for each of these algorithms. The simulation process is applied with the same setting for both algorithms, and each simulation is run with eight iterations. The difference

between generated plans of the same synthetic population is a result of the start-time random affectation steps.

Once activity-plans for all agents are generated, transportation is simulated using MATSim [17], which is a mesoscopic traffic simulator. The simulation requires three files: the demand (plans) file, which contains the generated plans of the whole population (126,151 agents); the catchment area multimodal network (84,429 nodes and 330,464 links); and the configuration file, which contains the simulation settings.

The output of the simulation is a multimodal dynamic traffic model; the simulation generates a set of files, for example, a plans file, which contains the simulated plans including their travel routes and the updated activity-setting, and an events file, which contains the trips of agents through the network at each time-step.

To evaluate the quality of the agent-plans, a (dis)utility function (score) $S$ measure is defined, which is composed of two functions: activity utility $Ua$ and trip (dis)utility $Ut$. The activity utility depends on the activity-setting (start-time, end-time, and duration), while the trip utility depends on its travel-mode. The score is calculated as the sum of all activity utilities $Ua_{act}$ plus the sum of all trip utilities $Ut_{trip}$. The score equation is defined in Equation (40).

$$S_{plan} = \sum_{act \in plan} Ua_{act} + \sum_{trip \in plan} Ut_{trip} \tag{40}$$

The activity utility $Ua$ is defined as the sum of five (dis)utility terms: utility of performing activity ($Ua_{dur,act}$), disutility of waiting time to begin activity ($Ua_{wait,act}$), disutility of late arrival ($Ua_{late.ar,act}$), disutility of early departure ($Ua_{early.dp,act}$), and disutility of too-short duration ($Ua_{short,act}$), which is the case when time spent by an agent is shorter than the minimum duration of the activity. The defined utility function is in Equation (41).

$$Ua_{act} = Ua_{dur,act} + Ua_{wait,act} + Ua_{late.ar,act} + Ua_{early.dp,act} + Ua_{short,act} \tag{41}$$

The trip disutility $Ut$ is defined as the sum of the constant of the trip transport mode ($C_{mode,trip}$) and disutility of travel time ($Ut_{trav\_time,trip}$), travel cost ($Ut_{trav\_cost,trip}$), travel distance ($Ut_{trav\_dist,trip}$), and public transport transfer ($Ut_{transfer,trip}$). The trip disutility function is then defined by Equation (42)

$$Ut_{trip} = C_{mode,trip} + Ut_{trav\_time,trip} + Ut_{trav\_cost,trip} + Ut_{trav\_dist,trip} + Ut_{transfer,trip} \tag{42}$$

For more details about the above utility functions, we refer the reader to [17].

A good plan must perform an activity with its typical duration in order to start the activity within the window (opening–closing) to avoid the waiting time or the late arrival penalties. For each plan-trip, the agent must minimize its travel time, cost, and distance. In the simulation step, at each iteration and for each agent, the executed plan receives a score value. Consequently, the agent receives two score values: a best score $BestScore$, which refers to its best plan score, and an average score $AvgScore$, which refers to the average score of all its plans.

To evaluate the quality of each population, two measures are defined: the average best (*avg.BEST*) represents the average score of the best plan of each agent $a$ from the population *popu* (see Equation (43)), and the average mean (*avg.AVG*) represents the average score for each agent $a$ (see Equation (44)).

After both scenarios (with and without time-constraints) were simulated, a comparative approach was applied, which compares both scenarios-plans in terms of the defined score values (*avg.BEST* and *avg.AVG*). The comparison of the approaches serves to evaluate the benefit (score) of using the time-constraints compared to these plans without constraints.

$$avg.BEST = \frac{1}{|popu|} \times \sum_{a \in popu} BestScore_a \tag{43}$$

$$avg.AVG = \frac{1}{|popu|} \times \sum_{a \in popu} AvgScore_a \qquad (44)$$

### 7.3. Model Parameters Setting

Figure 26 shows the average error per activity for each configuration for both end-time and duration. Globally, the different configuration has a similar pattern of end-time component; they show a very good result for the primary activities work and education, while for secondary activities, leisure, shopping, and restaurant, the results are still good with some limits in terms of quality. There is more variation in the duration-level from one configuration to another.

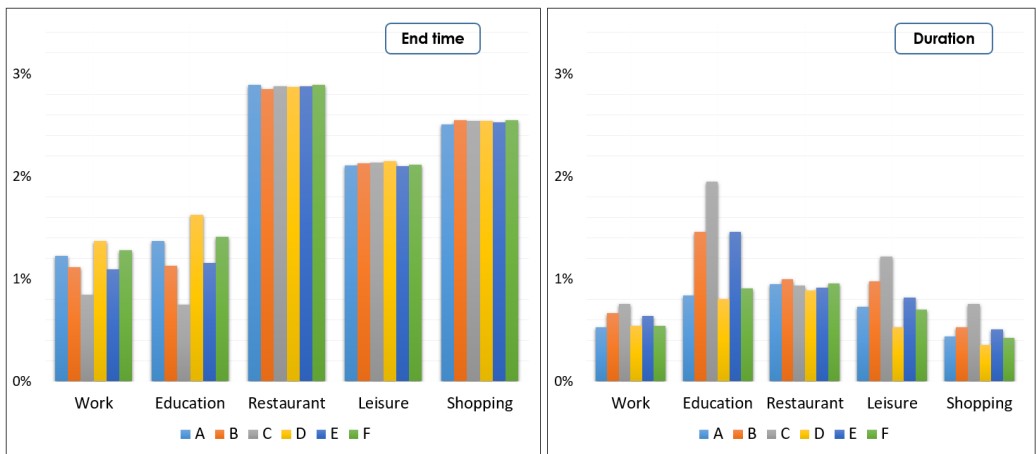

**Figure 26.** Comparison of time estimation error of the different configurations. (**Left**): at the end-time level, the configurations have almost the same activity-error value for restaurant, leisure, and shopping; the differences are at work and education activities, with an advantage belonging to (C), then to (E) and (B). (**Right**): at the duration-level, larger differences appear; configuration (D) is better than the others, followed by (A), (F), and (E). A compromise has to be made on these two criteria: configuration (E) shows the best trade-off between duration and end-time.

For better understating, the end-time results are first detailed, followed by the duration results. The configurations show almost the same results for the activities: leisure, shopping, and restaurant at the end-time. The main difference is then at the work and education, with an advantage of the configuration (C); this advantage is due the use of a maximum dynamic coefficient, which allows selecting an end-time from the largest possible range. The second on the list is the configurations (B) and (E), which have similar results. Then, (A), (F), and (D) have the worst results because, once the duration is selected, the end-time is deduced (no further end-time selection). At duration-level, the results are inverse compared with end-time; (D) is the best one, and (C) is the worst.

In conclusion, the selection of an end-time value depends on the way that the duration is selected. The configuration (E) shows a balance for both end-time and duration levels (second and fourth respectively). (E) and (F) use random dynamic coefficients, which are the basis of the proposed approach, with the advantage of (E) at the end-time level (which is the priority) compared with (F). (E) is selected to be the configuration of the remaining work.

### 7.4. Experimental Results

A comparison between the output of our model and the reference *ENTD* distributions is presented in Figures 27 and 28.

Figure 27 plots the distribution of durations: we can see, for example, that *education* occupation typically takes 3 h or 8 h, with two modes at these durations. The first mode represents students with half a day of lessons, and the second is the duration of courses for a full day at school. *Work* has two main duration modes: the first mode is around 4 h;

this mode represents an agent who works for a half of the day, while the second mode is between 8 and 9 h and represents an agent who performs a full day of work. *Leisure* has several modes: first, there are agents who spend a short leisure duration (5 to 15 min), mainly, for these agents, going for a walk and staying for a short period in the nearest park; the second part is agents who perform for 15 to 35 min and who go to the park in the morning or in the evening of a weekday; the remaining agents perform for a longer duration (more than 45 min) and are often parents who go to a larger park with their children, especially on the weekend. *Restaurant* has a typical duration of one hour.

Overall, activity-durations are well-fitted: for all activities the distribution of activities' duration output by the algorithm is similar to its reference *ENTD* distributions.

Figure 28 shows the time when an activity typically ends. According to the input census, employees usually finish work at noon if they work a half day or between 17 and 19 h for those working a full day. Teachers end at 12 h in the morning and 16 h 30 in the afternoon. Many leisure and shopping occupations end at 17 h. Shopping is more frequent in the morning, probably due to groceries errands. Lunch usually finishes by 13 h 30, while a dinner ends after 21 h.

Times of *Work* and *Education* are correctly modeled with some minor exceptions at peak hours, as the two modes' half-day and full-day are closely fitted. However, the approach has some difficulties fitting the input data of secondary activities:

- *Shopping* distribution is underestimated in the morning and at lunch time (from 9 h to 12 h), mainly because shopping trips to the bakery or at lunchtime are not considered. In the evening, the shopping time distribution is over-estimated: first, the construction of the activity-plan (where shopping is a secondary activity) is mainly attached to agents who perform a primary activity (work or education), which restricts the horizon time for shopping to only the evening time. Secondly, shopping has a large distribution time from 7 h to 22 h, which will give more freedom for agents to perform shopping later in the day, while in France, the large shopping areas close between 19 h to 21 h, and only small areas are still open until 22 h. At the national scale, part of the population does shopping activity once per week and in large shopping areas, while in the catchment area, the people have tend to shop in the local small areas. Restricting the closing time of shopping places to 19 or 20 h can considerably reduce the number of observations from ENTD (the proposed approach's training data), which will affect the performance of the proposed approach.
- *Leisure* distribution is overestimated at the end of the day due to the direct impact of performing the primary activities early in the day, which implies a delay in the start of leisure activity.
- *Restaurant* shows a major difference with the ENTD since the model dismisses of the lunch activity occurring at noon. The activity is shifted at the end of the day, and one can see the modeled ending time of dinner follows the same tendency as in ENTD but with more participation.

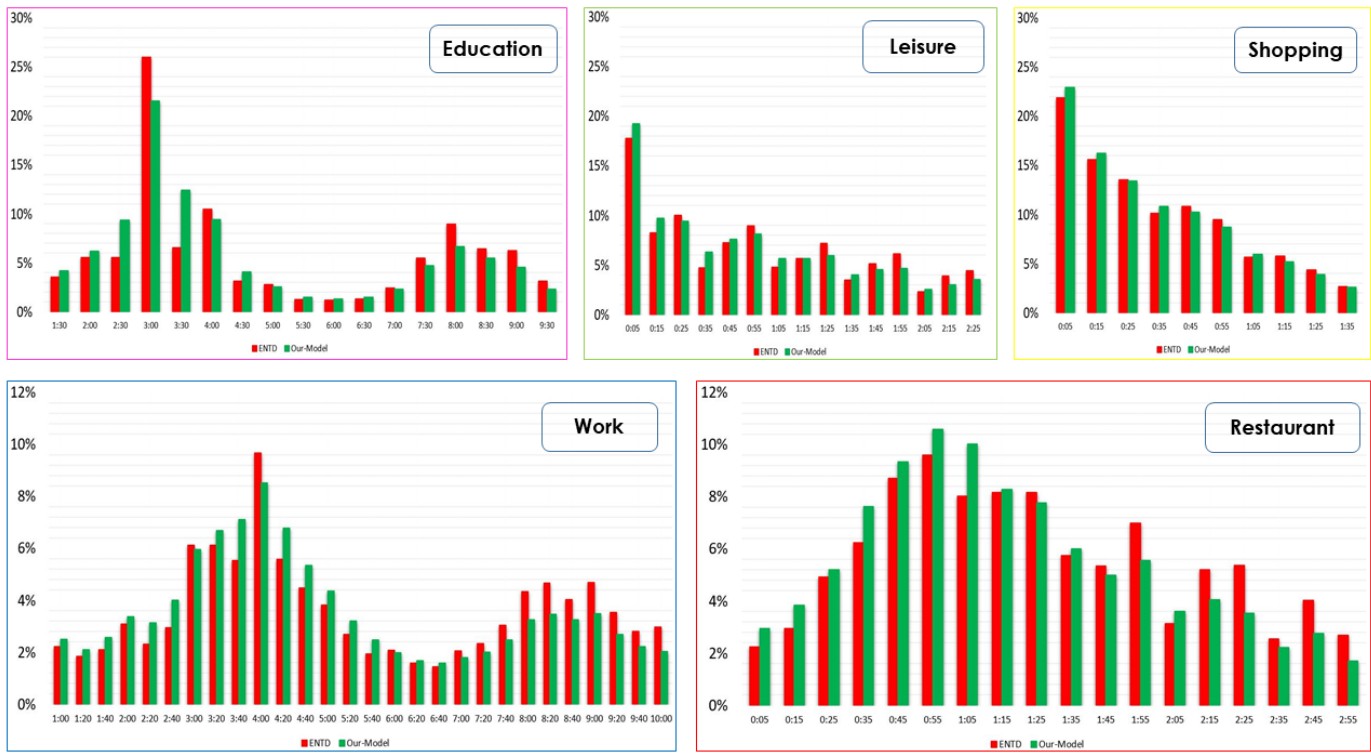

**Figure 27.** Activity-duration: a comparison of duration distributions between ENTD (red) and the proposed approach (green).

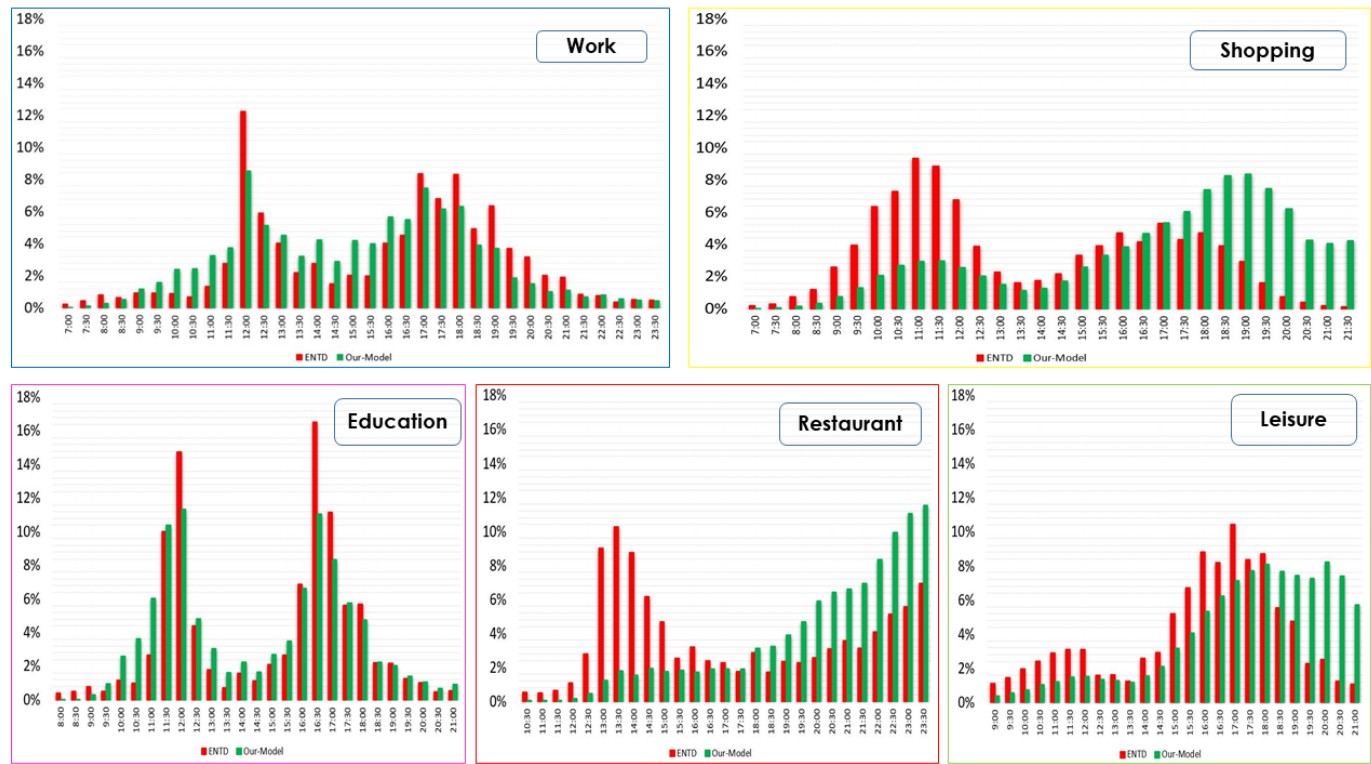

**Figure 28.** Activity end-time: a comparison of end-time distributions between ENTD (red) and the proposed model (green). Work and education have similar distributions to those of ENTD. Shopping and leisure have some differences in the distributions (due the region and the time of year). Restaurant-activity has two peaks (midday and evening) at ENTD, representing lunch and dinner, while in the proposed model, only the dinner is considered.

Another aspect to assess is the interest of the approach compared to the performance of the simulated plans with and without time-constraints in terms of score value.

Figure 29 presents the average score value after transportation simulation, for plans generated both with and without the constraints. To compare these two approaches, five generated set of plans are built, and their trips are simulated throughout the day independently. Chain of activities and their location are the same in each of the five generated populations, the difference being in the time of departure, which is a result of a random-dependant process.

The scores plotted in Figure 29 are the two average scores: mean plans score (avg.AVG) and best plans score of each scenario, averaged on five generated set of plans.

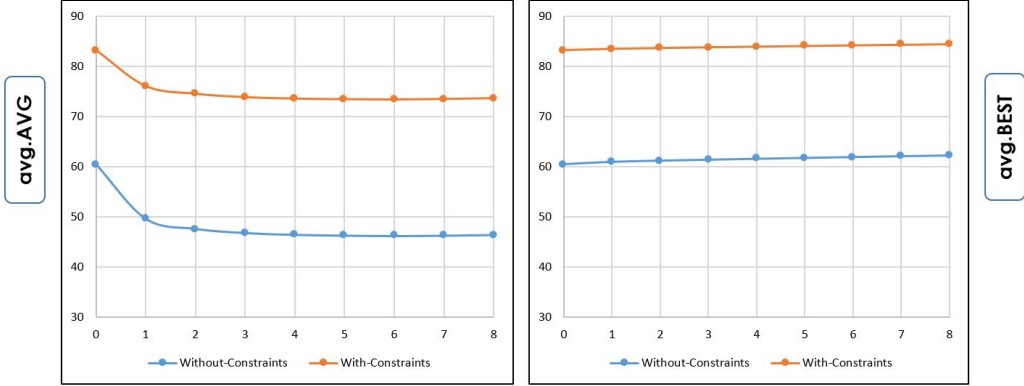

**Figure 29.** Evaluation of two score functions over the iterations (with and without constraints). Both scenarios (with/without constraints) follow the same pattern, and the proposed approach generates better plans comparable to those generated without constraints

In the first place, the average mean (avg.AVG) measure is analyzed for both scenarios. This measure passes by two phases: decreasing phase and improvement phase. At the first phase, the score values decrease due the generation of new plans (from the initial plan by changing its trip routes or its activity time settings); these plans can have a lower score (slower travel route due the changing of traffic or a shift in the activity-duration and starting time) compared to the best plans for the next iterations. In the second phase, the population's score improves very slowly as a result of the simulation of better new (updated) plans with faster routes and better activity time settings.

Secondly, the average best (avg.BEST) measure value is improved (slowly) for both scenarios over the iterations. The plans generated by the proposed approach improves their travel routes (traffic assignment), while the plans generated without constraints improve both their routes and their activity time settings. These plans are initially not good enough due the dismiss of the time-constraints.

In both measures, the plans generated with constraints lead to a higher score than those generated without constraints: it is mainly the consequence of a better-fitted initial population (in terms of activity time settings). By applying temporal constraints on the planned activities, the agent can perform the activity with a typical duration and will arrive (or leave) the activity-place within its opening time window, while plans generated without constraints risk having waiting time, arriving late, leaving early, or performing the activity with a shorter duration than the activity's minimum duration.

## 8. Discussion

The proposed approach fits well the duration of activities. For the activity time distribution, two patterns are distinguished: the primary activities work and education fit well. The output of these distributions is similar to the referenced distribution (in ENTD).

However, the proposed approach shows some difficulty fitting a set of secondary activities in some parts of the day. This drawback could be the result of multiple reasons: the primary activities are first inserted into the plan, followed by the secondary activities,

and the proposed approach assigns the duration and end-time for the activity by following their order in the activity-chain, which means that the time-setting of secondary activities depends mainly on the time-setting of the first activities; the second reason is that there was no consideration of shopping in the morning or at lunchtime, and the same is true for restaurant activity (at lunchtime), where the lunch activity is considered as an internal activity of work or education, providing a larger time-distribution for the secondary activities; the third reason is the difference between the national-scale and the catchment area, where restricting the time-window of activities will impact the fitting data and the performance of the approach.

The proposed approach guarantees the execution of the activities with typical duration within the time window, but it cannot guarantee that the following activities will have the same distribution of agents as the reference model (ENTD), generating an overestimation at the end of day for some activities, due the large freedom given to the agents to choose their start-time of the primary activities.

The simulation shows that these plans generated with time-constraints gave a better score compared to those generated without constraints. The main reason is the outcome of respecting these constraints, which allow agents to perform the plans with typical settings (duration, time window), which maximizes the utility (minimize the dis-utility) score functions value.

Even though the output model is coarse, we have found the method to reflect the mobility trends experienced in the department, and it outperforms an approach with no consideration of time-constraint.

Our algorithm is detailed and reproducible granted one can gather the same kind of temporal data on activities. It uses only public data and can therefore be directly applied for any case study on any French territory.

The algorithm was test in a pre-COVID-19 world. In a post-COVID-19 situation, especially work hours but also other activities can be expected to change with the introduction of remote work. The approach would be of particular interest to practitioners in a post-COVID-19 scenario once the new habits have settled or to test scenarios of curfew with different time constraint.

## 9. Conclusions

In this paper, we are interested in building a synthetic population of a suburban area with realistic activity-schedules. Typically, sparse census and econometric data are available for that type of urban population, rendering microeconomic foundations hard to achieve; therefore the region becomes unfit to utility-based approaches. Our method applies activity-hours distributions extracted from public census with a limited corpus, drawing the beginning time of a potential next activity based on the end-time of the previous ones and the remaining time for the next ones to be completed during the day and predicted travel times. We show that our method is able to construct plannings with chain of activities comprising work, education, shopping, leisure, restaurant, and kindergarten that fit adequately real-world time distributions. We believe that this paper is of interest especially for the practitioner of the field, giving thoughtful insights into the development of temporal aspects of activity-planning in population synthesis.

**Author Contributions:** Investigation, Y.D. and R.B.; Methodology, Y.D., R.B. and F.D.; Supervision, R.B.; Writing—original draft, Y.D. and R.B.; Writing—review and editing, R.B., F.D. and M.Z. All authors have read and agreed to the published version of the manuscript.

**Funding:** This research was funded by the E3S project, a partnership between Eiffage and the I-SITE FUTURE consortium. FUTURE bénéficie d'une aide de l'État gérée par l'Agence Nationale de la Recherche (ANR) au titre du programme d'Investissements d'Avenir (référence ANR-16-IDEX-0003) en complément des apports des établissements et partenaires impliqués. FUTURE benefits from France grant managed by the National Research Agency (ANR) under the Investments for the Future program (reference ANR-16-IDEX-0003) in addition to the contributions of the establishments and partners involved.

**Data Availability Statement:** ENTD datasets are available in: https://www.statistiques.devel oppement-durable.gouv.fr/enquete-nationale-transports-et-deplacements-entd-2008 (accessed on 8 July 2021).

**Conflicts of Interest:** The authors declare no conflict of interest.

## Abbreviations

The following abbreviations are used in this manuscript:

| | |
|---|---|
| ABM | Activity-Based Modeling |
| ALBATROSS | A Learning BAsed TRansportation Oriented Simulation System |
| CEMDAP | Comprehensive Econometric Microsimulator for Daily Activity-travel Patterns |
| ENTD | Enquête Nationale Transports et Déplacements |
| DEPLOC | Déplacements locaux |
| MATSim | Multi-Agent Transport Simulation |
| MORPC | Mid-Ohio Regional Planning Commission |
| SCHEDULER | Computational-process modelling of household activity scheduling |
| TASHA | Travel Activity Scheduler for Household Agents |

## Appendix A. Illustration Example

**Table A1.** Computed values for a {H1, K1, W, K2, H2, L, Sh} chain.

| *A* | *B* | A-End-Time $ed_A$ | Mode | Speed (m/s) | Travel Time $TT_{A \to B}$ | B-Start-Time $ST_B$ |
|---|---|---|---|---|---|---|
| H1 | K1 | 08:25:00 | car | 9.10 | 00:02:10 | 08:27:10 |
| K1 | work | 08:27:10 | car | 13.00 | 00:14:06 | 08:41:16 |
| work | K2 | 13:56:00 | car | 14.30 | 00:13:13 | 14:09:13 |
| K2 | home | 14:09:13 | car | 11.60 | 00:01:50 | 14:11:03 |
| H2 | leisure | 19:45:00 | pt | 4.30 | 00:09:00 | 19:54:00 |
| leisure | shopping | 21:18:15 | walk | 1.15 | 00:05:00 | 21:23:15 |
| shopping | H3 | 21:59:00 | pt | 6.50 | 00:12:12 | 22:11:12 |

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
