# Peer review of "Modeling Activity-Time to Build Realistic Plannings in Population Synthesis in a Suburban Area"

_applsci, doi:10.3390/app11167654_

Round 1

Reviewer 1 Report

Dear authors,

paper is well written with all elements of a scientific paper. The methodology and research results are well explained.

Here are several comments:

  1. National Transport and Travel Survey (Enquête Nationale Transports et Déplacements) ENTD is from 2008. Please, just briefly explain why this data is still applicable in 2021.
  2. Today most parents drive their children to primary (elementary) school. So some elements from kindergarten could also be applied for elementary schools. Please, comment.
  3. I suggest adding discussion in the paper, or expand conclusion with discussion.

Reviewer 2 Report

It is not clear the value added of the paper please better clarify.

The experimental results does not help in understanding the value added. It is difficult to state if the proxi of the real distributin is acceptable and if it depend on data generation at the beginning or other aspect of the algorithm.

while algorithms applied may help in identify some shift in the time choice of daily scheduling, they cannot fill the gap generated by absence of information like how many people go back home for lunch or how many goes shopping during the lunch break.

The paper should better identify what are the information needed to properly constraint the model

paragraph 5 and 6 are not straightforward.  reader risk to get lost: features of the dataset, assumptions employed and how they have been included in the algorithm. I suggest restructuring and keep the three aspect clearly separated

Similarly keep separated description of the data generated ( the syntetic polulation) and clearly specifiy when and how they differ from the original data (ENTD dataset) as well description of the simulation

Reviewer 3 Report

This is a very well-written manuscript regarding the development of an activity-based transport model for a suburban area of Paris using data from the National Transport and Travel Survey of France, using the Multi-Agent Transport Simulation package. I think that the manuscript will be very useful for the practitioners and researchers, as it explains thoroughly the process and the validation of the results, therefore I propose to be accepted in its present form.

Nevertheless, I have some minor comments for the authors that they could consider during the proofing of the manuscript (if accepted for publication):

  1. I would define all the acronyms.
  2. Line 88: I would use total instead of global, as global may be misunderstood for the population of the Earth.
  3. Lines 102, 106, and 110: I wouldn't start a sentence with a citation of this kind in this form. You may use the names of the authors and the citations can follow.
  4. Line 334: "considers" perhaps should be "considered".
  5. Perhaps I would discuss a little bit more what kind of applications and practical usefulness could the results have for other researchers.
